# End-to-end Deep Reinforcement Learning for Stochastic Multi-objective Optimization in C-VRPTW

**Abdo Abouelrous**                                                                 *a.g.m.abouelrous@tue.nl*
*Department of Information Systems, Faculty of Industrial Engineering and Innovation Sciences, Technical University Eindhoven, The Netherlands*

**Laurens Bliek**                                                                          *l.bliek@tue.nl*
*Department of Information Systems, Faculty of Industrial Engineering and Innovation Sciences, Technical University Eindhoven, The Netherlands*

**Yaoxin Wu**                                                                               *y.wu2@tue.nl*
*Department of Information Systems, Faculty of Industrial Engineering and Innovation Sciences, Technical University Eindhoven, The Netherlands*

**Yingqian Zhang**                                                                       *yqzhang@tue.nl*
*Department of Information Systems, Faculty of Industrial Engineering and Innovation Sciences, Technical University Eindhoven, The Netherlands*

**Reviewed on OpenReview:** *https://openreview.net/forum?id=Wwtb1tYnp5&noteId=mBj1k9IWWf*

## Abstract

In this work, we consider learning-based applications in routing to solve a Vehicle Routing variant characterized by stochasticity and multiple objectives. Such problems are representative of practical settings where decision-makers have to deal with uncertainty in the operational environment as well as multiple conflicting objectives due to different stakeholders. We specifically consider travel time uncertainty. We also consider two objectives, total travel time and route makespan, that jointly target operational efficiency and labor regulations on shift length, although more/different objectives could be incorporated. Learning-based methods offer earnest computational advantages as they can repeatedly solve problems with limited interference from the decision-maker. We specifically focus on end-to-end deep learning models that leverage the attention mechanism and multiple solution trajectories. These models have seen several successful applications in routing problems. However, since travel times are not a direct input to these models due to the large dimensions of the travel time matrix, accounting for uncertainty is a challenge, especially in the presence of multiple objectives. In turn, we propose a model that simultaneously addresses stochasticity and multi-objectivity and provide a refined training mechanism for this model through scenario clustering to reduce training time. Our results show that our model is capable of constructing a Pareto Front of good quality within acceptable run times compared to three baselines. We also provide two ablation studies to assess our model's suitability in different settings.

**Keywords:** Stochastic, Multi-Objective Optimization, Vehicle Routing Problem, Reinforcement Learning, Active Search, End-to-End

## 1 Introduction

The Vehicle Routing Problem (VRP) is a problem of significant industrial relevance in contemporary economies where vehicle delivery operations play a pivotal role in the supply chain (Vidal et al., 2020). Real-life variants are often characterized by certain challenging features. Examples of such features include

stochasticity and the presence of several (conflicting) objectives simultaneously. Accounting for these features during decision-making is important. Otherwise, sub-optimal decisions may lead to elevated costs. In practical settings, computational budgets are often limited as solutions need to be generated quickly in accordance with operational requirements (Horvitz, 2013). To that end, one seeks optimization techniques that are capable of delivering near-optimal solutions efficiently in the presence of computational challenges posed by factors like uncertainty and multi-objectivity.

In the literature, numerous techniques have been proposed to solve Combinatorial Optimization (CO) problems like VRP, ranging from hand-crafted heuristics to Machine Learning (ML) models that are independently able to generate solutions - also known as end-to-end methods (Kotary et al., 2021). The spectrum also covers hybrid methods, which incorporate both heuristics and ML models. ML models are particularly useful in cases where problem parameters follow a known distribution. In such cases, an ML model can be trained on a dataset of problems from this distribution and used to generate solutions for problem instances arising in the future. This is because ML models have the capacity to learn from the collective expert knowledge. Through learning common solution structures and relationships to problem parameters. Although the training resources required are not trivial, ML models can be quickly applied for solving thereafter. (Bengio et al., 2021; Giuffrida et al., 2022; Mazyavkina et al., 2021; Zhang et al., 2021; Kool et al., 2018).

In our estimation, CO problems treated in the literature by ML barely treated stochasticity and multi-objectivity together. The objective of our study is to extend the framework of ML in optimization to settings that jointly combine both problem features. Solving a multi-objective problem requires the definition of a Pareto Front (Ngatchou et al., 2005), comprising of a set of Pareto optimal solutions by which improving one objective can not be done without worsening other objectives. Solutions would then have to be validated on a sample of scenario realizations to estimate their quality and feasibility. In an optimization context, this poses a challenge that requires novel intervention. This is because multi-objectivity induces the generation of several solutions to estimate the Pareto Front, while stochasticity imposes that each solution be feasible with a good score for a large number of stochastic realizations. The repeated evaluation of multiple solutions, in this setting, is a cumbersome task that requires careful choices in the optimization methodology to maintain computational costs within practical limits.

In response, we present a framework that addresses the given computational issues in this paper to solve Capacitated Vehicle Routing Problem with Time Windows (C-VRPTW), a popular variant of VRP (Liu et al., 2023). We specifically focus on stochasticity in travel times and consider two objectives that are travel time-based, although our method could account for more different objectives. The major contributions prescribed by this paper are:

- Presenting the first end-to-end (independently constructs solution without interference of a heuristic) deep-learning model that jointly treats stochasticity and multiple objectives for routing.
- Providing a retraining mechanism for parameter uncertainty through an active search that considers multiple objectives rather than just one objective.
- Presenting a scenario-clustering technique that enhances the computational efficiency of model retraining during active search.

The remainder of the paper is organized as follows. Section 2 introduces previous research. Section 3 describes the relevant problem in detail. Section 4 outlines our methodology. Section 5 presents numerical experiments and results. Lastly, Section 6 asserts the conclusions.

## 2 Previous Work

We restrict our attention to ML applications in stochastic and/or multi-objective cases for routing. These models generally rely on concepts of Reinforcement Learning (RL) to make decisions. These approaches fall into two major categories, hybrid and end-to-end. Hybrid models integrate a ML model with a heuristic to enhance the heuristic's performance. The way in which the ML model is applied largely depends on the heuristic at hand. A common example is that the RL agent selects a heuristic with some probability from a set of discrete heuristics. In contrast, End-to-end models are able to independently solve a routing problem

without interference of a heuristic. When deciding on the ordering of node visits, the decision to visit a node $j$ after node $i$ is based on a probability $p_{ij}$. Such a probability may be determined by features of nodes $i$ and $j$, the status of the current (partial) solution and the estimated corresponding reward. In both categories, the decisions are determined by interaction of the problem input with the ML model's parameters which are optimized during model training to maximize the final reward.

For the multi-objective setting, works like Wu et al. (2024) and Deng et al. (2024) explore applications to routing with hybrid methods through genetic algorithms. Yao et al. (2017), on the other hand, focus on hyper-heuristic selection by the RL agent to complete a solution. In the Stochastic setting, Bayliss (2021) make use of simulation-augmented optimization for urban-routing. Joe & Lau (2020) employ a hybrid approach with a genetic algorithm to address a dynamic problem with stochastic customers. Works integrating both stochastic and multi-objective components are rare, at least in routing. Tozer et al. (2017) propose an RL model that selects voting methods based on social choice theory for path-finding. Niu et al. (2024) and Niu et al. (2021) embed a decision tree that learns node orderings in a genetic algorithm in location and vehicle routing, while Peng et al. (2023) adapt the same approach but for multi-modal transportation. Zhang et al. (2024), in contrast, optimize multi-modal transportation routing using a framework aided by simple Q-learning.

Recent innovations in the literature have prompted the use of Deep Reinforcement Learning (DRL) to solve CO problems. Among the earliest of end-to-end ML applications in routing was Kool et al. (2018) who leverage a Graph Attention network Veličković et al. (2017) to estimate $p_{ij}$. Kwon et al. (2020) extended on this by presenting a model that uses multiple solution trajectories during model training and solution inference (generation) which we refer to as Policy Optimization with Multiple Optima (POMO) for simplicity. POMO is able to process complex graph information and greedily decide on the next node visit most likely to minimize the objective value(s). The model is well-credited for its ability to generate solutions of high quality in a relatively short time compared to alternative solvers and heuristics for a multitude of routing problems given a certain distribution. During training, POMO learns the relationship between node orderings and the resulting final objective values to look for orderings that minimize the objective in the future.

The impressive performance of DRL-based methods like POMO has led to adaptations in multi-objective and stochastic cases for routing. For multi-objective CO routing problems, DRL has been proposed in works like Sarker et al. (2020), Wang et al. (2023), Chen et al. (2023) and Lin et al. (2022). For applications in stochastic CO routing problems, works like Schmitt-Ulms et al. (2022), Achamrah (2024), Iklassov et al. (2024) and Zhou et al. (2023) showcase end-to-end methods addressing demand and travel-time uncertainty, which is usually accounted for by fine-tuning the pre-trained model's parameters. In these works, routing problems are solved in an end-to-end fashion whereby the RL model decides on successive node visits. This is in contrast to other DRL innovations that rely on hybrid methods (Jia et al., 2025; Son et al., 2024) for stochastic and multi-objective applications.

To the best of our knowledge, there is no end-to-end DRL mechanism for solving routing (or CO) problems characterized by both uncertainty and multiple objectives. Many of the multi-objective DRL architectures can not be easily adapted to account for uncertainty and vice-versa. On the other end, existing hybrid methods depend on the heuristics at hand which are often problem-specific. For example, deriving voting methods or learning node orderings with decision trees may not be suitable for more complicated vehicle routing variants such as ones with time windows. Ideally, one would like to come up with an end-to-end model that can be easily adapted to solve a multitude of routing problems without the complexities associated with external assumptions or hyper-heuristics. The novelty of this paper is embedded in deriving such a framework for solving C-VRPTW, although the framework can be easily adopted to solve other routing problems as explained below.

## 3 Problem Description

Consider a C-VRPTW instance with parameter input $\mathcal{P}$ and stochastic travel times $\mathcal{T}$, where $\mathcal{T} \in \mathcal{P}$. Furthermore, assume the problem is characterized by a set of $K$ objectives for which preference $\lambda_k$ corresponds to objective $f_k(.) \forall k \in K$. The preferences $\lambda_k$ are embedded in a vector $\lambda$. For simplicity, assume $\sum_{k \in K} \lambda_k = 1$ and $0 \leq f_k(.)$.

Such preferences could reflect the decision-makers view of importance of one objective relative to another. Such preferences may not be explicitly defined initially, but repeatedly solving the problem with different $\lambda_k$ values enables the decision-maker to understand the relationship between decisions and corresponding objective values through the construction of a Pareto Front. Thereafter, the decision-maker can choose decisions on the Pareto Front that align with their preferences. The proposed model should therefore be able to approximate a Pareto optimal solution for any set of preferences as much as possible.

Let $n$ be the number of nodes that may be visited in a solution and $V$ be the set of nodes, so that $|V| = n$. Each node $i \in V$ is characterized by some features that depend on the problem at hand, such as demand or time-windows. Between each pair of nodes, there is an arc with travel times corresponding to the stochastic parameters $\mathcal{T}$. That said, $\mathcal{T}$ is a $n \times n$ matrix with elements $t_{ij}$ representing the travel times between nodes $i$ and $j \in V$. Let $t_{ij}$ follow some predetermined probability distribution $f_{ij}(.)$. Depending on the problem, a depot may be defined as well. If so, we index the depot by 0, so that $0 \in V$ and define stochastic travel times $t_{0i}$ and $t_{i0}$ for each other node $i \in V$.

The objective is to jointly optimize all the $K$ objectives while satisfying operational constraints. In C-VRPTW, time-window constraints, as well as objective values, are largely determined by travel times. As such, we are dealing with a CO problem where the objective values and constraint satisfaction for a set of actions/decisions are uncertain.

Our proposed model strives to define a mapping - specified by a set of parameters $\theta$ - between graph information and associated node ordering in a solution that maximizes rewards (i.e. delivers good objective values). Our model takes into account the uncertainty in travel times represented by the graph's arc lengths and conflicting rewards (objectives). The mechanism by which we define this mapping serves the purpose of our study. Furthermore, cases with uncertain travel times $\mathcal{T}$ pose a greater challenge than others in the literature such as in Niu et al. (2021) which deal with stochastic demand. This is because the number of travel time parameters is much more than demand parameters, invoking a larger computational capacity to process the associated uncertainty. Furthermore, different node orderings can give different feasibility and solution scores for a given set of travel time realizations, in contrast to the case with stochastic demand. In the following section, we present a method that treats the problem presented above.

## 4 Methodology

We propose a single machine learning model whose training is partitioned into two phases. One phase deals with the multi-objective component of the problem and the latter with the stochastic component. Our model is a variant of the POMO model whose parameters correspond to $\theta$. It is firstly trained on a sample of multi-objective deterministic instances. Afterwards, the parameters embedding information on the travel times are re-trained to account for the stochasticity. The retraining is done by means of an Efficient Active Search (EAS) which we will explain below. Finally, given problem input $\mathcal{P}$ and a vector of preferences $\lambda$, we apply a solution generation procedure with the re-trained POMO model to produce a solution. The Pareto Front can then be estimated by repeatedly solving the problem $\mathcal{P}$ for different values of $\lambda$. An overview of our overall method is illustrated in Figure 1. In the following section, we elaborate on the components of our approach.

### 4.1 Multi-Objective Component

The standard POMO model seen in Kwon et al. (2020) defines a mapping with parameters $\theta$ between problem parameter input $\mathcal{P}$ and a resulting reward/objective value. More precisely, the model's main architecture is given by an encoder-decoder duo. The encoder with parameters $\theta_{enc}$ takes as input problem parameters to construct an embedding $\omega$, which is a summary of the problem features as explained in Kool et al. (2018). $\omega$ is then used alongside other dynamic information pertaining to the current status of the solution by the decoder with parameters $\theta_{dec}$ to take actions until a solution is complete with a corresponding reward. Since the reward is constant for every set of parameter input $\mathcal{P}$, the decoder can automatically process $\omega$ and make a decision.

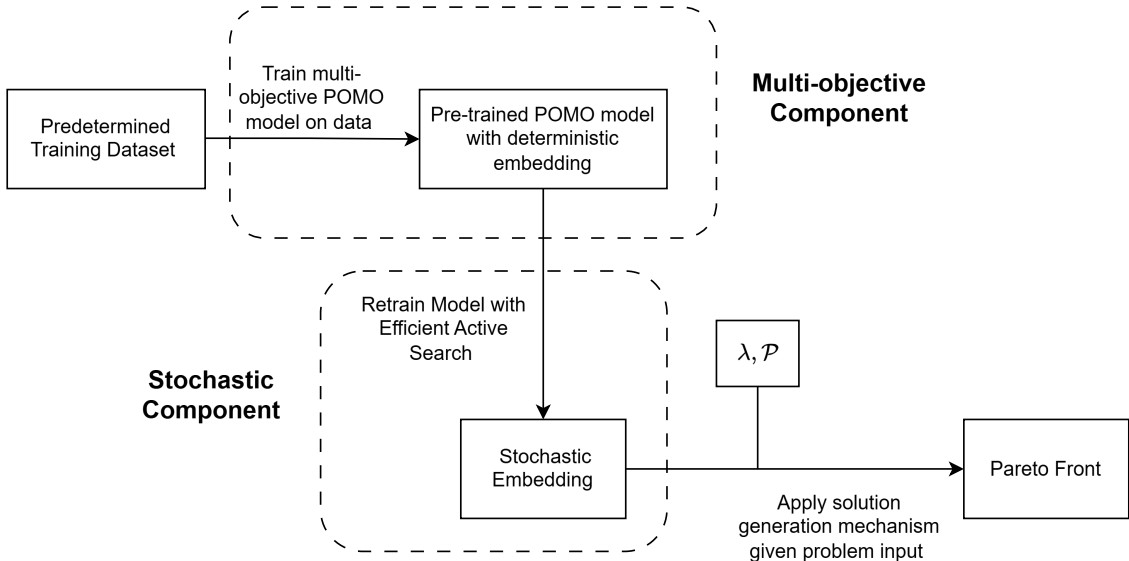

Figure 1: Visual illustration of our complete approach using pre-trained POMO model.

In the multi-objective setting, the reward depends on the preferences $\lambda$. To that end, the information needed to make a decision is not entirely prescribed by the embedding $\omega$. In turn, Lin et al. (2022) propose a multi-objective variant of POMO. The idea is to equip the decoder with a Multi-Layer Perceptron with parameters $\psi$ that takes $\lambda$ as input and determines the corresponding decoder parameters $\theta(\lambda|\psi)$. The resulting POMO model can then be used to construct a Pareto Front by repeatedly sampling realizations of $\lambda$ and solving the corresponding problem for a single realization one at a time. Since POMO models can solve a single instance quite efficiently, the Pareto Front would not require a significant computational budget to be constructed.

Figure 2 depicts the process by which a solution is generated with the model of Lin et al. (2022). This model and its training correspond to the Multi-objective component in Figure 1. In the following section, we explain how the embedding $\omega$ generated from this model is updated by means of an EAS to adapt to stochastic travel times $\mathcal{T} \in \mathcal{P}$.

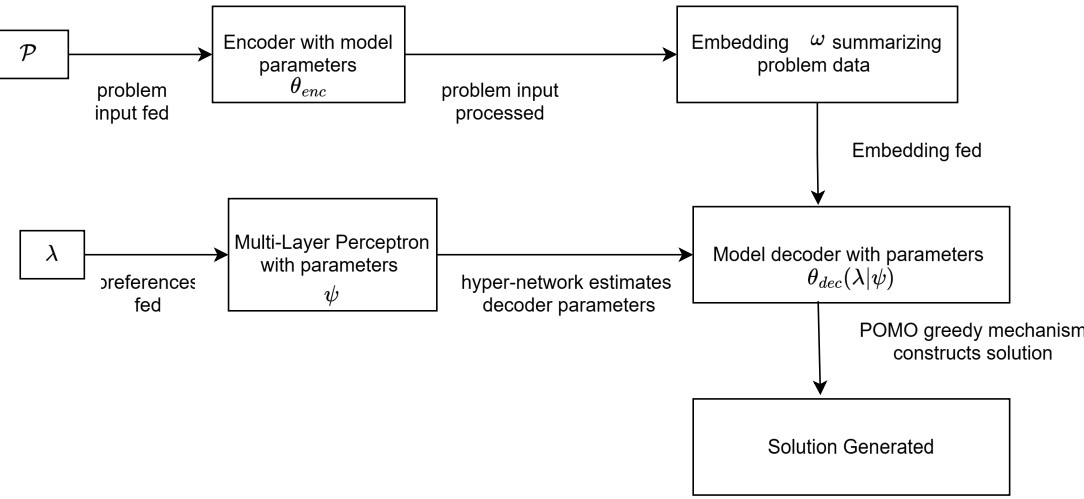

Figure 2: Overview of solution generation with the model of Lin et al. (2022).

To compute the reward for a given solution $\pi_i$ for problem instance $s_i$, Lin et al. (2022) propose a linear aggregation technique. The $K$ objectives are aggregated into a single objective (reward) as follows:

$$\sum_{k \in K} \lambda_k f_k(\pi_i) \tag{1}$$

with $f_k(.)$ representing objective $k$. The term in (1) corresponds to the reward $L(\pi_i | \lambda, s_i)$ for solution $\pi_i$ given instance $s_i$ and preferences $\lambda$. Solution $\pi_i$ is generated according to a probability distribution $p_{\theta(\lambda)}(\pi_i | s_i)$ which can be constructed in a greedy or sampling fashion.

There is an important consideration with regards to linear aggregation. While linear aggregation has been proposed in studies like Lin et al. (2022), it is unable to produce a non-convex front, which may arise in CO problems based on mixed integer programs like C-VRPTW (Pappas et al., 2021). One should, thus, consider the observed objective values and see whether they are in line with the comparison metrics.

The model is trained by repeatedly sampling a preference vector $\lambda$ and a corresponding batch of $B$ instances. Thereafter, $M$ different solution trajectories are created where each trajectory represents one possible solution. The average reward over all the $M$ different solution trajectories and $B$ instances is computed, where the reward from one trajectory corresponds to some aggregation of the $K$ objectives such as in (1). The average reward is, in turn, used to compute the loss function and associated gradient for optimizing the model's parameters $\theta$. To estimate the gradient loss, we make use of the REINFORCE training algorithm of Williams (1992) with the ADAM optimizer. The training algorithm is summarized in Algorithm 1.

---

**Algorithm 1** Training POMO Model from Lin et al. (2022)

---

1: **Input**: preference distribution $\Lambda$, instances distribution $\mathcal{S}$, number of Training steps $T$, number of objectives $K$, batch size $B$, number of solution trajectories $M$.
2: **Output**: model parameter $\theta$.
3: **for** $t = 1 \cdots T$ **do**
4:     Sample $\lambda_k$ from $\Lambda$ for $k \in K$.
5:     Sample $B$ instances from $\mathcal{S}$
6:     Generate $M$ different solutions using $p_{\theta(\lambda_k)}(.|s_i)$ for each instance $s_i$.
7:     Define shared baseline reward for instance $s_i$ using $b_i = \frac{1}{M} \sum_{j=1}^{M} L(\pi_i^j | \lambda, s_i)$ $\qquad \forall i \{1, \cdots, B\}$
8:     Compute gradient $\Delta_\theta = \frac{1}{BM} \sum_{j=1}^{B} \sum_{i=1}^{M} [(L(\pi_i^j | \lambda, s_i) - b_i) \Delta_{\theta(\lambda)} \log p_{\theta(\lambda)}(\pi_i^j | s_i)]$
9:     $\theta = ADAM(\theta, \Delta_\theta)$
10: **end for**
11: **return** $\theta$.

---

### 4.2 Stochastic Component

Given that travel times are not encoded with the problem input for POMO, but implicitly represented by a parametrized embedding $\omega$, an EAS is introduced in Schmitt-Ulms et al. (2022) by which the model is re-trained to adapt to the different possible stochastic realizations of $\mathcal{T}$. In the retraining phase, only $\omega$ is updated. In each step, a gradient $\Delta_\omega$ is computed with $M$ solution trajectories by the following formula:

$$\Delta_\omega = \frac{1}{M} \sum_{j=1}^{M} [(L(\pi^j | \lambda, s) - b) \Delta_{\omega(\lambda)} \log p_{\omega(\lambda)}(\pi^j | s)] \tag{2}$$

and $\omega$ is updated by means of gradient ascent. For consistency, we use the ADAM optimizer for the update. Observe that the gradient computation in (2) is similar to the one in Line 8 in Algorithm 1 - based on the REINFORCE algorithm. The major differences is that this update concerns only one instance rather than a batch of $B$ instances, and the only parameters being updated are $\omega$. As such, the probability distribution $p_{\omega(\lambda)}(\pi^j | s)$ is determined by updates in $\omega$.

The embedding is updated over a series of $T_\omega$ steps for instance $s$ with expected travel times $E[\mathcal{T}]$ and then evaluated on $W$ stochastic realizations every $t_e$ iterations. $t_e$ can be interpreted as the evaluation

frequency. Further, the number $W$ should be sufficiently large to address many possible realizations. Since this evaluation is expensive, it is only done in a few of the $T_\omega$ steps. The $\omega$ value that gives the lowest mean aggregate objective value over all $W$ scenarios in the evaluation steps is selected as the final embedding.

There is an important consideration regarding the active search described in Schmitt-Ulms et al. (2022). In the evaluation epoch, the resulting embedding $\omega$ is evaluated on a large number of realizations $W$. Many of these realizations are similar and the resulting objective values as specified by the pretrained POMO model from Section 4.1, rendering these evaluations somewhat repetitive and unnecessary. Ideally, we are interested in evaluations that are distinctive in input and resulting objective values. Secondly, because $W$ is rather large, this constrains the number of evaluations we can carry out during retraining. That said, a more efficient evaluation method may not only reduce retraining time but also allow us to evaluate the embedding more frequently with a smaller $t_e$, giving us a larger search space that may possibly result in a larger embedding.

To account for the aforementioned considerations, we propose some adjustments to the EAS. Firstly, we sample the $W$ possible scenarios. Thereafter, we cluster these methods to get a subset of scenarios that are sufficiently representative of the spectrum of realizations. To do that, we use the clustering method of Abouelrous et al. (2022).

We consider the initial embedding $\omega$ generated by training the entire model in Algorithm 1. We solve, the $W$ problems in batches of $B$ to speed up their evaluation. For each instance $s_i$, we select the solution trajectory from the $M$ trajectories with the highest aggregate reward $L(\pi^*|s_i) = \max_{j \in \{1, \cdots, M\}} L(\pi^j|s_i)$ which will be used to compare with other scenarios. The highest is chosen instead of the mean as it represents the best solution found. The aggregate reward is calculated with fixed preferences $\hat{\lambda}$ during clustering, although the preferences may change later during the active search. Fixing parameter values and assuming perfect information has been shown to be very beneficial for clustering (Abouelrous et al., 2022). Two scenarios $s_i$ and $s_k$ are clustered together if:

$$|L(\pi^*|s_i) - L(\pi^*|s_k)| < \epsilon \tag{3}$$

with $|.|$ being the absolute operator and $\epsilon$ being some threshold. Thus, scenarios $s_i$ and $s_k$ are clustered together if their objective values with the initial embedding are sufficiently close.

The first scenario automatically forms a cluster. The scenarios are compared to the clusters in order. So, scenario $s_k$ is compared to the scenario representing cluster 1. If it meets (3), they are clustered together. Otherwise, $s_k$ is compared to the scenario in cluster 2 and so forth. If $s_K$ is compared with all existing clusters and not clustered with any, it forms a new cluster. The result of the clustering procedure is a subset of $\overline{W} < W$ scenarios. These scenarios are then used in the embedding evaluations which happen every $t_e$ iterations. Since $\overline{W}$ is much smaller than $W$, we can opt for a smaller $t_e$ value and evaluate the embeddings more frequently and efficiently.

Our active search is summarized in Algorithm 2. An important distinction with the active search proposed in Schmitt-Ulms et al. (2022) is that preferences $\lambda$ are repeatedly sampled during re-training to ensure that the uncertainty is incorporated under different sets of preferences determined by the decision-maker. When new preferences are sampled, the decoder parameters have to be updated, but one should ensure that the embedding $\omega$ is not altered as it is being retrained (Line 14). Similarly, during evaluation (Line 20), all decoder parameters are updated except $\omega$. We also fix the preference vector $\lambda$ during evaluation (Line 20) to ensure a fair evaluation as the model's performance for different values of $\lambda$ may be uneven.

The returned $\omega^*$ is then saved to solve the problem instance with unknown travel times using a greedy policy that maximizes the probability of the next visited node. The distribution of nodes to be visited is defined by the retrained POMO model. We refer to the model trained using Algorithm 2 as **EAS-cluster**.

---

**Algorithm 2** EAS-cluster Training

---

1: **Input**: number of search steps $T_\omega$, total number of realizations to consider $W$, clustering threshold $\epsilon$, batch size during evaluation $H$, initial preferences $\hat{\lambda}$, evaluation frequency $t_e$.
2: **Output**: updated embedding $\omega^*$
3: **Initialization**: initial embedding $\omega$
4: **for** the $W$ scenarios **do**
5:      Sample them in batches of $H$
6:      Evaluate the $H$ instances using $\omega$ and fixed preferences $\hat{\lambda}$.
7:      Cluster the instances according to (3)
8: **end for**
9: $\overline{W} =$ Nr. of clusters.
10: $\omega^* = \omega$
11: Evaluate $\omega^*$ on the $\overline{W}$ scenarios with $\hat{\lambda}$ to determine initial mean aggregate reward $L(\pi^*|\overline{W}, \omega^*)$
12: **for** $t = 1 \cdots T_\omega$ **do**
13:      Sample new preferences $\lambda$
14:      Update Decoder parameters without updating $\omega$
15:      Solve deterministic instance using current embedding $\omega^*$ for given $\lambda$.
16:      Compute gradient $\Delta_\omega$ using (2).
17:      $\omega = ADAM(\Delta_\omega, \omega)$
18:      **if** $t$ is an evaluation epoch (a multiple of $t_e$) **then**
19:          Evaluate $\omega$ on the $\overline{W}$ scenarios with $\hat{\lambda}$. Let $L(\pi^*|\overline{W}, \omega)$ be the resulting mean aggregate reward. Ensure that $\omega$ is not updated in the evaluation.;
20:          **if** $L(\pi^*|\overline{W}, \omega) < L(\pi^*|\overline{W}, \omega^*)$ **then**
21:              $\omega^* = \omega$
22:              $L(\pi^*|\overline{W}, \omega^*) = L(\pi^*|\overline{W}, \omega)$
23:          **end if**
24:      **end if**
25: **end for**
26: **return** $\omega^*$.

---

### 4.3 Extension to other DRL Architectures

Our proposed framework is largely based on the POMO architecture common to both the multi-objective and stochastic component. In that sense, our framework is largely restricted to POMO-based architectures. However, the architecture is largely considered a pillar in using DRL for solving CO problems (Wang et al., 2024). Much of the recent literature on multi-objective DRL work mentioned in Section 2 also makes use of encoder-decoder architecture of Lin et al. (2022) making the framework relevant to many applications.

Extension to other DRL architectures is not straightforward. Assuming another encoder-decoder architecture where the decoder parameters contain a graph embedding $\omega$, the general framework may be still applicable. Figure 1 offers an important guideline, however, into solving stochastic multi-objective problems. One should first establish the multi-objective component before proceeding with the stochastic one. The reason being that the model should first be able to generate multiple diverse solutions for a given problem before being able to assess these solutions on a set of stochastic scenarios. This is particularly evident in the retraining mechanism in Section 4.2 whereby preference vectors $\lambda$ are repeatedly sampled during the EAS in Algorithm 2.

## 5 Numerical Experiments

To evaluate our method, we propose a series of numerical experiments where we solve a bi-objective ($K = 2$) C-VRPTW with stochastic travel times. We jointly minimize the objective of total travel times and makespan (total time consumed by longest route). We compare the performance of our method on unseen instances relative to alternative methods. For the comparison, we consider the final objective value as well as the total

run time. In the following sections, we elaborate on the set-up of our numerical experiments and analyze their results.

## 5.1   Set Up

We consider three instance classes of sizes $n \in \{50, 100, 200\}$ with $B = 20$ instances for each $n$. For all instance classes, the locations are given in 2-D space where the x-y coordinates are sampled from a square of length 1. The vehicle capacities are 40, 50 and 70 for the three classes in increasing $n$. Node demand is sampled uniformly as integer from the interval $[1, 9]$. Time windows are such that the lower time window $tw_{low}$ is sampled uniformly as integer from the interval [0,16], the time window width $tw_{width}$ from the interval [2,8] and the upper time window equal to $\min\{tw_{low} + tw_{width}, tw_{horizon}\}$ with the $tw_{horizon}$ being the planning horizon. Naturally, the depot's time window is $[0, tw_{horizon}]$, and we set $tw_{horizon}$ to 18. Service times are sampled uniformly between [0.2,0.5]. Lastly, the mean travel times $E[\mathcal{T}]$ are equal to the euclidean distance, and the travel times follow a normal distribution with standard deviation $0.2 \times E[t_{ij}]$ for arc $(i, j)$.

For each instance size $n$, we train a different POMO model with 200 epochs and 100,000 episodes, giving a total of 20,000,000 instances. The instances were sampled from the aforementioned distribution in batches of 64 for $n = 50$ and 32 for $n = 100, 200$. Training was conducted on a GPU node with 2 Intel Xeon Platinum 8360Y (Intel (2025)) Processors and a NVIDIA A100 Accelerator (Nvidia (2025)). The training times of the multi-objective POMO model were 18, 60 and 144 hours for $n =$50, 100 and 200.

For EAS-cluster, we initially consider a sample of $W = 1,000$ scenarios in batches of $H = 64$ from which we decide on the clusters $\overline{W}$. In doing so, we consider initial preferences $\hat{\lambda}_1 = \hat{\lambda}_2 = 0.5$ to assign all objectives equal importance when reducing scenarios. Two scenarios are clustered together if their aggregated objective values $\hat{\lambda}_1 f_1(\pi^*) + \hat{\lambda}_2 f_2(\pi^*)$ differ by less than $\epsilon = 1\%$ where $\pi^*$ is the best solution found by the pre-trained ML model from Section 4.1 before EAS for the corresponding instance.

For $n =$50, 100 and 200, EAS-cluster produced 7.3, 6.5 and 5.75 clusters on average across the $B = 20$ instances. This shows that we can significantly reduce the number of scenarios as many of these correspond to similar figures of $f_1(.)$ and $f_2(.)$ for fixed $\hat{\lambda}$. Furthermore, the number of clusters deceases as $n$ increases, likely due to differences in objective values becoming less significant as objective values like travel time become larger for larger $n$.

Once the scenarios are clustered, we conduct an active search with $T_\omega = 2,500$ steps where the embedding is evaluated every $t_e = 100$ steps. We use three benchmark methods, that test different aspects of our study. They are as follows:

- **NoEAS:** the model without active search, which is simply the multi-objective POMO model from Lin et al. (2022) without any updates to embedding $\omega$ applied to solve the deterministic instance. This baseline has been repeatedly shown to outperform other methods such as NSGA-II (Kalyanmoy, 2002) and MOEA/D Zhang & Li (2007) in solving multi-objective combinatorial optimization problems, especially in routing, so we focus on comparing with it. We generate 101 values of $\lambda$, where $\lambda_1$ is computed from 100 evenly spaced intervals in the range [0,1] and $\lambda_2 = 1 - \lambda_1$ to represent the Pareto Front.

- **EAS-basic:** with the active search of Schmitt-Ulms et al. (2022) with $t_e = 250$. Similarly, we use the same 100 evenly spaced $\lambda$ realizations as above. Solution inference is done using a greedy policy defined by the retrained model.

- **LGA:** the method of Niu et al. (2021). It is a learning-based Genetic Algorithm which makes use of a decision-tree to learn optimal customer orderings. We adjust this method slightly to allow for time-window configurations, since it was originally developed for a VRP variant with stochastic demand only. Adding time-windows increases the frequency of infeasible solutions. We respond to this by increasing the number of Genetic Algorithm iterations to 2,000 - from the original 200 - and the number of evaluation scenarios in the population to 50 - from the original 10. LGA required no pre-training, although a new ML supervised model had to be trained for each instance during solution generation.

- **LGA+noML**: An LGA variant where no machine learning component is used. Solution search is done purely using genetic perturbations (operators).

- **LALNS**: An adapted version of LGA where the random genetic operators are replaced by an Adaptive Large Neighborhood Search (ALNS) - similar to Pisinger & Ropke (2007) - which is known to work well with C-VRPTW. The ALNS is combined with the decision tree classifier from LGA. We run it for 200 iterations due to the high computational cost of ALNS.

The resulting solutions for each method and preference $\lambda$ are evaluated on $R = 500$ stochastic realizations of the travel times. The resulting mean objective value is used for comparison as it represents the expected objective value at the Pareto Front for the given $\lambda$. Infeasible realizations are excluded from the evaluation. Solution Inference was done on the same GPU nodes used for training, (Intel, 2025; Nvidia, 2025). For LGA variants, no GPU is needed, so an AMD EPYC 9654 (AMD, 2025) CPU node was used. The results of the baselines compared to our method on the proposed set of instances are given in the following section.

## 5.2 Results

To compare results, we consider the same hyper-volume technique mentioned in Lin et al. (2022). The hyper-volume measures the area covered by the Pareto Front from reference point, with larger hyper-volumes translating to better Pareto Fronts. Let $\mathcal{P}$ define the Pareto Front for a certain policy and $r^*$ be some reference point that is dominated by all solutions in $\mathcal{P}$. Then, the hyper-volume $HV(\mathcal{P})$ of $\mathcal{P}$ is given by the volume of area $\mathcal{S}$ that is defined as follows:

$$\mathcal{S} = \{r \in \mathbb{R}^k | \exists y \in \mathcal{P} \text{ such that } y \prec r \prec r^*\} \tag{4}$$

where $y \prec r$ indicates that solution $y$ dominates solution $r$. Given a baseline $l$ and instance $b$, the associated hyper-volume is given by $HV(\mathcal{P}^b)_l$. To estimate this volume, we make use of the Python package 'pymoo' (Blank, 2026). We are particularly interested in the percentage increase in hyper-volume due to using EAS-cluster. This percentage is then averaged over all $B$ instances in the test set to give our main performance measure $Z$ that is given by:

$$Z = \frac{1}{B} \sum_{b=1}^{B} \frac{[HV(\mathcal{P}^b)_{EAS-cluster} - HV(\mathcal{P}^b)_l] \times 100}{HV(\mathcal{P}^b)_l} \tag{5}$$

For EAS-cluster and EAS-basic the retraining times of the active search are compared. We report the average ratio $tf$ of EAS-cluster's active search $t_{av}^{EAS-cluster}$ relative to the active search of EAS-basic $t_{av}^{basic}$. More precisely:

$$tf_{av} = \frac{t_{av}^{EAS-cluster}}{t_{av}^{EAS-basic}} \tag{6}$$

such that values $< 1$ indicate a smaller run time with EAS-cluster. Furthermore, we report the total run time for solution inference per instance as $t_{inf}$, which represents the total time it would take to generate and evaluate for a given number of solutions. For EAS-cluster and EAS-basic, this includes the total run time with the active search and solution generation using POMO's greedy policy for all $\lambda$ values, while only the latter is incorporated for NoEAS as it lacks active search. For each of the POMO-based methods, inference times are consistent among all $B$ instances for fixed $n$. For LGA and LALNS, this includes the solution generation and training of the supervised learning model which happens during inference.

We also report the number of solutions $Q_r$ generated at the Pareto Front by each method. Note that this number is equal for all POMO-based models (101). Larger $Q_r$ values indicate a method's ability to generate more solutions at the Pareto Front, and possibly, increased diversity. Lastly, we report the average number of feasible realizations $R^f$ out of the total $R = 500$ for the solution constructed from the greedy policy for each $n$. This is particularly relevant to assess solutions' practicality as solutions which are feasible for only a handful scenarios might not be appealing even if they result in better objectives. Table 1 compares the result of EAS-cluster with the five baselines. Additional statistics on the percentage increase in hyper-volume are reported in Table 5 in Appendix C.

| n | Metric | EAS-cluster | EAS-basic | NoEAS | LGA | LGA+noML | LALNS |
|---|--------|-------------|-----------|-------|-----|----------|-------|
| 50 | $Z$ (avg. ratio) | 0 | -0.02% | 8.14% | 368.31% | 336.52% | 8.36% |
| | $tf_{av}$ (ratio) | 1 | 0.44 | $--$ | $--$ | $--$ | $--$ |
| | $t_{inf}$ (mins) | 49 | 58 | 42 | $\approx 0.17$ | $\approx 0.17$ | $\approx 2$ |
| | $Q_r$ (Nr.) | 101 | 101 | 101 | 8 | 8 | 18 |
| | $R^f$ (Nr./500) | 474 | 494 | 494 | 426 | 411 | 440 |
| 100 | $Z$ | 0 | -0.28% | 6.34% | 636.57% | 612.50% | 14.52% |
| | $tf_{av}$ | 1 | 0.43 | $--$ | $--$ | | $--$ |
| | $t_{inf}$ | 90 | 105 | 78 | $\approx 0.67$ | $\approx 0.67$ | $\approx 10$ |
| | $Q_r$ | 101 | 101 | 101 | 10 | 11 | 30 |
| | $R^f$ | 438 | 443 | 447 | 380 | 314 | 345 |
| 200 | $Z$ | 0 | -0.51% | 4.73% | 970.48% | 1041.67% | $--$ |
| | $tf_{av}$ | 1 | 0.40 | $--$ | $--$ | $--$ | $--$ |
| | $t_{inf}$ | 164 | 195 | 143 | $\approx 3.25$ | $\approx 3.25$ | $--$ |
| | $Q_r$ | 101 | 101 | 101 | 13 | 14 | $--$ |
| | $R^f$ | 422 | 430 | 410 | 257 | 263 | $--$ |

Table 1: Results EAS-cluster compared to EAS-basic on 20 different multi-objective C-VRPTW instances with stochastic travel times $\mathcal{T}$ for each instance size $n$.

For EAS-cluster, 7, 12 and 21 minutes per instance were spent on the EAS alone for the given values of $n$. For all values of $n$ (-0.02%,-0.28% and -0.51%), EAS-cluster results in a slightly worse Pareto Front than EAS-basic as given by values of $Z$ close to zeros. This slight deterioration could be due to the bias imposed by the evaluation on the few scenarios in $\overline{W}$. Statistical testing implies that the results are not significantly different from 0, with p-values of 0.98, 0.75 and 0.34 for sizes 50, 100 and 200 assuming a t-test with $B - 1 = 19$ degrees of freedom. To emphasize on the distribution of the value of $Z$ among the $B = 20$ instances, we provide Figure 3 in Appendix B. EAS-basic incurs a larger computational during the active search, consuming more than twice the run time of EAS-cluster for all instances as result of the larger number of evaluations on scenarios sampled from $W$. This gives $tf_{av}$ ratios of 0.44, 0.43 and 0.40 for the corresponding values of $n$, despite EAS-cluster incurring more than twice the evaluations of EAS-basic (25 to 10) during the 2,500 retraining steps. The resulting run times for EAS-basic are 58, 105 and 195 minutes per instance compared to EAS-cluster's 49, 90 and 164 minutes. Feasibility also slightly improves with EAS-basic as the number of feasible evaluations $R^f$ is 494 443 and 430 compared to EAS cluster's 474, 438 and 422 which could be attributed to the aforementioned bias. With these results, we observe that the reduction in computation time due to EAS-cluster outweighs the small deterioration in the Pareto Front hyper-volume and scenario feasibility.

The added value of active search, and EAS-cluster specifically, is shown in the significant improvement in hyper-volume compared to NoEAS, with $Z$ values of 8.14%, 6.34% and 4.73% for the given $n$. For the sake of clarity, we provide the distribution of the $Z$ values in Figure 4 in Appendix B. We see that $Z$ decreases as $n$ increases due to the increased difficulty of solving larger problems. The run times of NoEAS only include the time for Pareto Front construction which is similar for EAS-cluster, giving 42, 78 and 143 minutes. Solution feasibility is better for $n = 50$ and 100, with $R^f = 494$ and 447 scenarios. However, for $n = 200$, $R^f$ decreases to 410. This could be possibly due to the more pronounced effect of stochasticity in larger instances that can not be easily treated by the original embedding $\omega$. In fact, $R^f$ decreases as $n$ increases for all methods as it becomes increasingly difficult for a given solution to meet all the time windows of the $n$ customers.

We consider standard $LGA$ from Niu et al. (2021). Since the existing implementation we found makes use of multi-threading, it was difficult to accurately estimate the run time per instance, which we observed to be less than a minute. However, LGA gives significantly inferior solutions compared to EAS-cluster as given by $Z$ values of 368.31%, 636.57% and 970.48%. This can largely be attributed to the method's random genetic operators which can not easily produce feasible solutions in the case of time-windows. Furthermore, the simple decision tree is unable to learn information on time-windows from customer orderings, thus struggling

to find quality feasible solutions. A prominent advantage of POMO-based models, on the other hand, is their ability to learn complex CO problems which can be easily configured in the model's environment and adequately processed by the encoder. LGA is also unable to find as many solutions as POMO-based methods on the Pareto Front with 8, 10 and 13 solutions on average with respect to the 101 of the POMO-based models. Furthermore, since these solution do not correspond to a specific balance of objectives $f_k(.)$, it is difficult to verify their diversity for different decision-making criteria. In contrast, POMO-based methods can provide such diversity through $\lambda$ parameters to balance the objectives $f_k(.)$.

To further investigate the computational difficulty posed by time-windows for LGA, we consider the results of LGA+noML where the decision tree is omitted and solutions are generated purely by means of a genetic algorithm's perturbation operators. The performance is rather similar to standard LGA with hyper-volumes of 336.52%, 612.50% and 1041.67% for sizes $n =$ 50, 100 and 200. The number of feasible runs and solutions on the Pareto Front is fairly similar, except in the case of 100 where the number of feasible runs decrease to 314. These results stress the need for a heuristic suitable for solving a VRP variant with time-windows like ALNS.

Finally, we consider LALNS. As expected, the hyper-volume improves significantly compared to the two LGA baselines, although it still falls short to EAS-cluster. The mean hyper-volume increase is 8.36% and 14.52% for $n = 50$ and 100. The number of solutions observed at the Pareto Front was 18 and 30 respectively with 440 and 345 feasible evaluation scenarios. The run-times were also relatively short at 2 and 10 minutes approximately. For $n = 200$, we encountered an out-of-memory error as LGA's multi-threading mechanism and the application of ALNS at multiple points at the Pareto Front led to significant computational memory consumption. While these results indicate an improvement over LGA, LALNS is still inferior to other POMO-based methods which can treat complex CO problems although at a much higher run-time. Further, by observing the trend from $n = 50$ to 100, we expect that the difference in hyper-volume compared to EAS-cluster is significantly larger for larger instances. This asserts the added value of EAS-cluster in solving larger problems, which we investigate further in Section A.3.

For a clear comparison of the resulting Pareto Front of all the methods, we refer to Figure 5 in Appendix B. The figure compares the Pareto Front from EAS-cluster with each of the baselines for one of the test instances of size 100. The comparison with EAS-basic in Figure 5a shows that the Pareto Fronts are fairly close, while the comparison with NoEAS in Figure 5b shows a more prominent difference especially when preferences are more oriented towards minimizing travel-times. Lastly, the comparisons with LGA and LALNS in Figures 5c and Figure 5d demonstrate the superiority EAS-cluster's Pareto Front compared to either baselines. In Appendix A, we provide a series of ablation studies concerning the model's sampling policy and generalization to more variable scenarios and larger problem instances.

## 6 Conclusions

In this paper, we provided an end-to-end method to solve a multi-objective stochastic C-VRPTW. We specifically considered stochastic travel times and minimized total travel distance and make-span. Our model is based on a variant of POMO that is first trained on deterministic multi-objective instance and then retrained through an active search to account for travel time stochasticity. The model could not only be configured to treat a variety of routing problems through its flexible environment, but also account for more different objectives. The end-to-end mechanism of the model not only simplifies solution generation, but is also invariant of other external assumptions on the model environment and the action space.

We provide a training mechanism for the model and enhance it using a clustering algorithm that speeds up evaluations during active search and run time as a result. Our results show that our model is very close in performance compared to a baseline employing standard active search from the literature while being faster. It also significantly outperformed another POMO baseline that disregards stochasticity as well as a common learning-based method for VRP from the literature, resulting in more dominant Pareto Fronts.

Our model is able to generate as many solutions as desired for any set of preferences and in reasonable run times. Its greedy policy is also sufficient to generate good results, reducing the need for sampling policies often associated with POMO-based models. It also generalizes well to other instances with different

distribution than the one encountered during active search. Nonetheless, it often results in less feasible solutions compared to the considered baselines, possibly due to the bias induced by the clustered sample of scenarios. Furthermore, there still lacks a common framework that streamlines evaluation for specific cases like larger scale instance and non-convex Pareto Fronts.

Future research should focus more on synchronizing multi-objective training with active search and the derivation of superior clustering techniques while streamlining evaluation methods by which the associated model performance could be accurately judged. Additionally, problems with more objectives are certainly of interest.

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

| n | Metric | EAS-cluster (Monte Carlo Simulation) | EAS-cluster (Greedy) |
|---|---|---|---|
| 50 | $Z$ (avg. ratio) | 0 | 0.33% |
| | $t_{inf}$ (mins) | 17 | 49 |
| | $Q_r$ (Nr.) | 11 | 101 |
| | $R^f$ (Nr./500) | 489 | 474 |
| 100 | $Z$ | 0 | -1.26% |
| | $t_{inf}$ | 37 | 90 |
| | $Q_r$ | 11 | 101 |
| | $R^f$ | 470 | 438 |

Table 2: Results of EAS-cluster with Monte Carlo Simulation compared to greedy policy for each instance size $n$.

## A    Ablation Studies

To study the ability of our method to deliver improved results and generalize to settings with different distributions, we propose two sets of experiments. In the first set of experiments, we combine our method with Monte Carlo Simulation. This is in line with the recommendations in Schmitt-Ulms et al. (2022) who state that Monte Carlo Simulation may improve over a greedy policy. In the second set of experiments, we apply our method to solve a series of instances from a different distribution with more variable travel times.

### A.1    Monte Carlo Simulation

With the final embedding $\omega^*$, Schmitt-Ulms et al. (2022) use Monte Carlo simulation to evaluate the different actions at every step of the solution generation, i.e. node visit. More precisely, at each step, the top 5 recommended actions are evaluated by applying them in the succeeding step and completing each of the 5 solution trajectories using the trained model's policy. For each step, the process is repeated for a fixed number of runs $ms_r$ and the average is taken to determine the best action in the next step.

We make use of the same Monte Carlo simulation technique. There are some adjustments, however, we ought to incorporate in the application of the Monte Carlo Simulation for VRP. Firstly, Schmitt-Ulms et al. (2022) consider a large number of simulation runs $ms_r$ - going up to 100s - for each problem instance. This results in long running times per solution as demonstrated by their numerical experiments with instances of size 200 going up to 48 minutes. Since we repeatedly have to apply this procedure for several realizations of $\lambda$ as each realization of $\lambda$ invokes a different solution, we consider a smaller value $ms_r = 10$ so that the total run time per instance falls within the same order of magnitude of around 1 hour. Furthermore, we limit the number of sampled $\lambda$ values to 11 instead of 101.

Unlike the orienteering problem in Schmitt-Ulms et al. (2022), it is not possible to realize the stochastic travel times after taking a step since repeated visits to the depot in VRP would then imply that we can travel back in time. As such, the travel time realization we make after taking a decision concerns the expected value of the travel time, while stochasticity is incorporated in future decisions by the Monte Carlo runs. To impose feasibility constraints we require that the decisions considered at each step are feasible for at least half of the $ms_r = 10$ runs, otherwise they are discarded. We require that solutions are only feasible 50% of the time. Similar to the greedy policy, we evaluate the resulting solution on a sample of $R = 500$ realizations and use the average to estimate the objective values for given $\lambda$ at the Pareto Front.

We compare the performance of the Monte Carlo approach to the greedy policy of EAS-cluster from Section 5.2 in Table 2. Additional statistics on the percentage increase in hyper-volume are reported in Table 6 in Appendix C. Due to the expensive evaluation, we only consider $n =$ 50 and 100. The reported $t_{inf}$ times also include the run times for active search which is identical for both methods as they used the same embedding.

The results generally imply that Monte Carlo simulation is of limited added value to multi-objective optimization. While a marginal improvement of 0.33% is observed for $n = 50$, a deterioration of 1.26% is observed for $n = 100$ compared to using the greedy policy. The average run time per instance as given by

| n | Metric | EAS-cluster (Greedy) | NoEAS |
|---|---|---|---|
| 50 | $Z$ (avg. ratio) | 0 | 8.39% |
| | $R^f$ (Nr./500) | 448 | 480 |
| 100 | $Z$ | 0 | 6.56% |
| | $R^f$ | 391 | 413 |
| 200 | $Z$ | 0 | 4.35% |
| | $R^f$ | 334 | 340 |

Table 3: Results of EAS-cluster with greedy policy compared to NoEAS and more variable travel times for each instance size $n$.

$t_{inf}$ is slightly less than a third of the greedy policy's run time, with Monte Carlo simulation requiring on average 17 and 37 minutes per instance for $n = 50$ and 100. Yet, the number of solutions on the Pareto Front generated from the greedy policy is 10 times as much.

Nonetheless, the Monte Carlo simulation slightly improves the number of feasible realizations from 474 to 489 and from 438 to 470 for $n$ =50 and 100. This could be attributed to the repetitive evaluations in the Monte Carlo runs that require that a node visit be feasible for at least half the $ms_r = 10$ for it to be considered in the solution. As a result, a stronger feasibility requirement is imposed compared to the greedy policy and more scenarios are satisfied by the resulting solution. For cases where feasibility is crucial, one might still resort to using Monte Carlo Simulation rather than a greedy policy.

## A.2 Travel Time Distribution

In this section, we study the performance of our retrained model EAS-cluster on a dataset with a different travel time distribution. We refrain from conducting an active search and simply run the greedy policy on the new dataset to test model generalization. For the new dataset, we consider standard deviation $E[t_{ij}] \times 0.4$ which is twice as variable as the travel times considered in Section 5.2.

As a benchmark, we consider NoEAS again. The rationale being that if the difference of $Z$ is still positive, then the model with updated embedding $\omega^*$ generalizes well to a reasonable extent beyond a model where no retraining is involved. We consider the greedy policy with $Q_r = 101$ solutions again. Since solution inference times $t_{inf}$ and number of Pareto Front points $Q_r$ are identical, we only report $Z$ and $R^f$ in Table 3. Additional statistics on the percentage increase in hyper-volume are reported in Table 7 in Appendix C.

The results are largely in line with those in Section 5.2. EAS-cluster improves the hyper-volume by 8.39%, 6.56% and 4.35% on average for $n = 50$, 100 and 200. While the performance of EAS-cluster may deteriorate slightly in a more variable dataset, it still outperforms NoEAS whose embedding $\omega$, is significantly 'out-of-tune' with the more variable travel times $\mathcal{T}$. NoEAS, however, provides more feasible solutions, giving $R^f = 480$, 413 and 340 solutions compared to EAS-cluster's 448, 391 and 334 for given $n$. This could be explained by the increased bias of cluster evaluations in a different dataset, although the difference in $R^f$ values decreases with larger $n$ due to the increased difficulty of solving larger instances as explained in Section 5.2.

## A.3 Larger-Scale Instances

In this section, we test the generalization of our method to larger instance classes than the ones upon which the model was trained. In particular, we consider instances of size $n$ =400 and 600 with vehicle capacities 100 and 120. We consider the original standard deviation of $E[t_{ij}] \times 0.2$ . For these experiments, we use the POMO model trained on instances of size $n$ =200. Due to the associated computational burden, we consider $B = 10$ instances for each size with 21 $\lambda$ values. This poses certain evaluation challenges as there is not a sufficient number of solutions on $\mathcal{P}$ to give a relatively accurate estimation of the hyper-volume using (5) (Lin et al., 2022), in contrast to the previous experiments with 101 solutions. Indeed, we saw that our method ended up scoring worse hyper-volumes despite consistently generating better objective values for almost all values of $\lambda$.

In response, we concluded that an alternative evaluation measures ought to be used which we take to be the average percentage decrease in mean euclidean distance $Z_{euc}$ between each point on $\mathcal{P}$ and the reference point $r^*$. We saw that this measures correlated positively with the definition of $Z$ as per (5) and could accordingly be used for this evaluation. Intuitively, it represent the average improvement offered by solutions in $\mathcal{P}$ over reference point $r^*$. In line with Section A.2, we compare with NoEAS and report the $Z_{euc}$ and $R^f$ metrics in Table 4.

The results demonstrate the consistent added value of EAS-cluster with respect to NoEAS even for larger instances where $Z_{euc}$ increased by 15.28% and 18.16% for $n = 400$ and 600. The number of feasible realizations deteriorated significantly for both methods to less than 300 and 200 for both instance sizes. This is because it becomes more difficult to find a solution that satisfies all time windows as the number of customer grows with highly variable travel times. Nonetheless, our method still yields significant improvements in the objectives for the considered instances. To assert this, we consider Figure 6 in B. We see that the distribution of $Z_{euc}$ among the different instances of size 600 in 6a is always above 10%. Furthermore, the distribution of Euclidean distances between points in $\mathcal{P}$ and $r^*$ for both EAS-cluster and NoEAS is given in 6b and 6c for one of the instances of size 600, such that the former is skewed towards larger values indicating larger improvements over $r^*$ by the corresponding solutions in $\mathcal{P}$.

| n | Metric | EAS-cluster (Greedy) | NoEAS |
|---|---|---|---|
| 400 | $Z_{euc}$ (avg. ratio) | 0 | 15.28% |
|  | $R^f$ (Nr./500) | 288 | 295 |
| 600 | $Z_{euc}$ | 0 | 18.16% |
|  | $R^f$ | 110 | 188 |

Table 4: Results of EAS-cluster with greedy policy compared to NoEAS for larger instances.

# B Figures

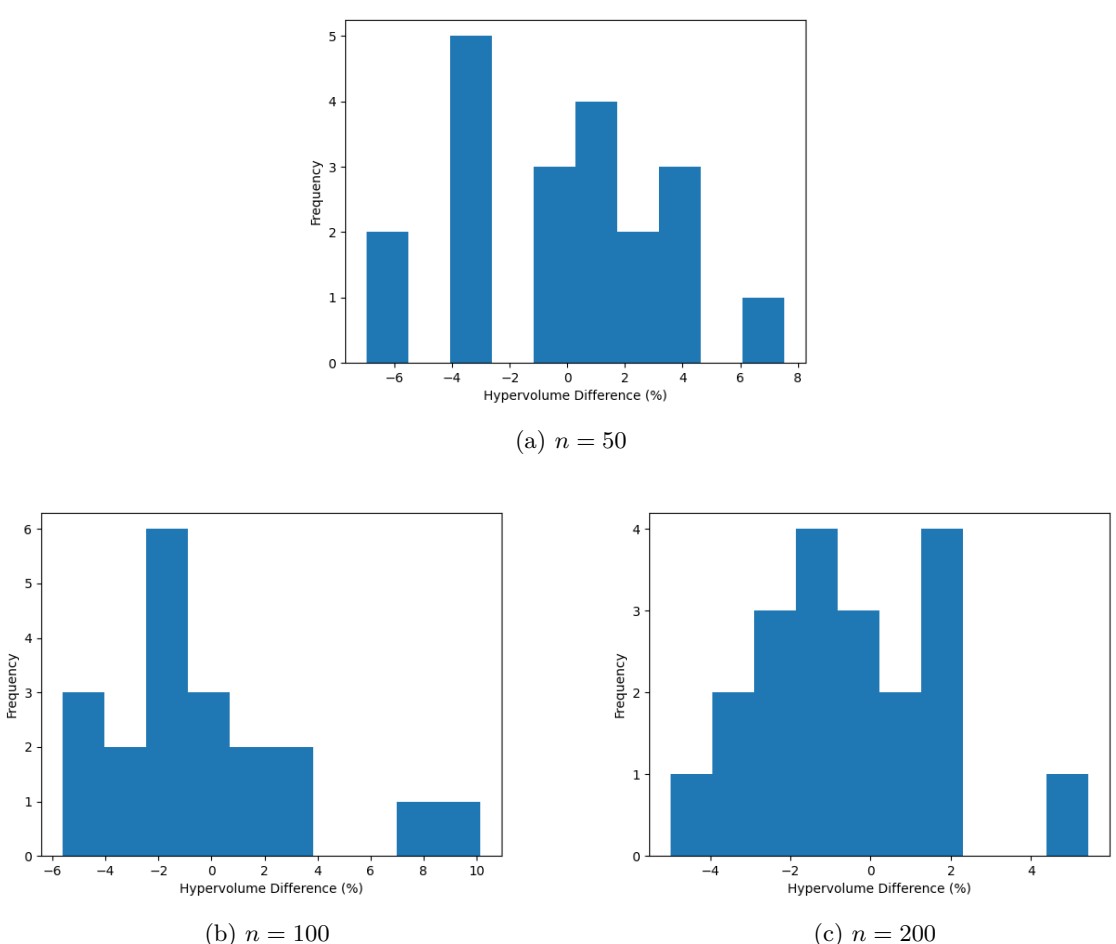

(a) $n = 50$

(b) $n = 100$

(c) $n = 200$

Figure 3: Histogram of $Z$ values of EAS-cluster against EAS-basic.

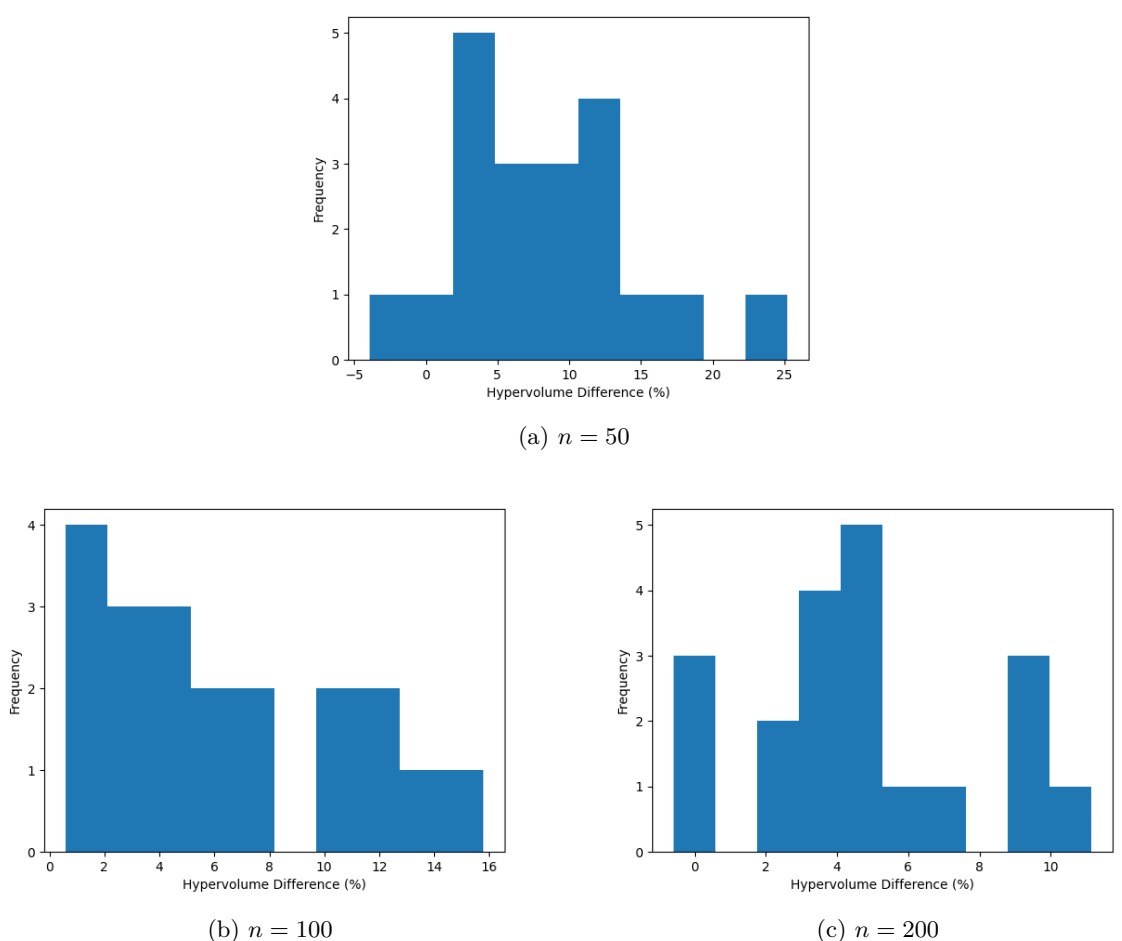

(a) $n = 50$

(b) $n = 100$

(c) $n = 200$

Figure 4: Histogram of $Z$ values of EAS-cluster against NoEAS

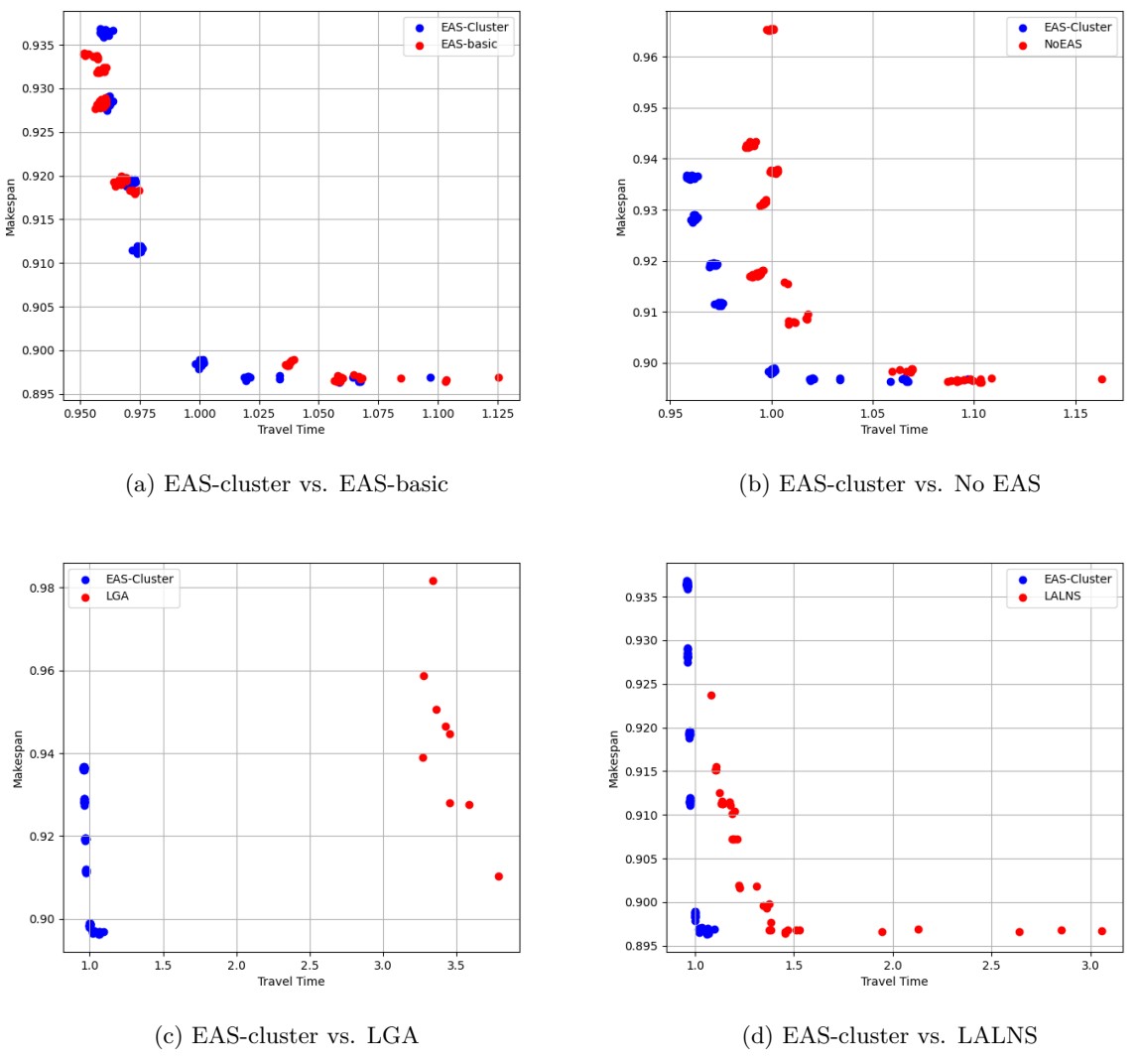

(a) EAS-cluster vs. EAS-basic

(b) EAS-cluster vs. No EAS

(c) EAS-cluster vs. LGA

(d) EAS-cluster vs. LALNS

Figure 5: Plots comparing the Pareto Front generated by EAS-cluster against other methods for one of the instances of size 100.

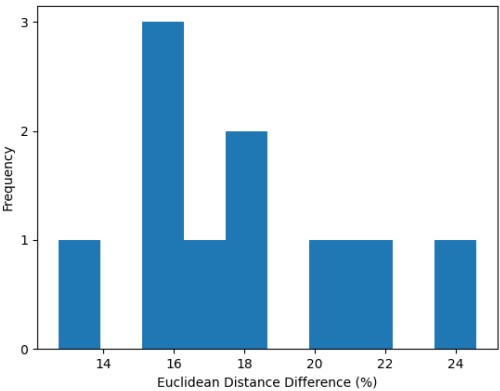

(a) Distribution of mean euclidean distance $Z_{euc}$ among the different $B = 10$ instances of size 600.

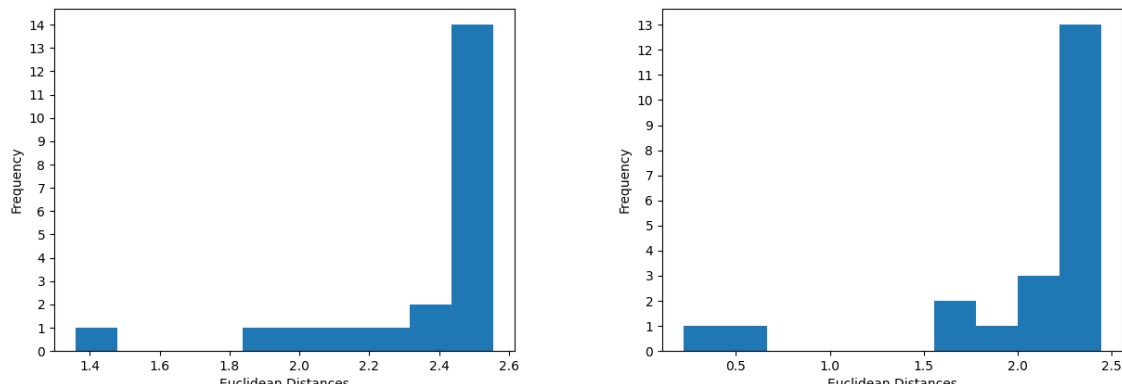

(b) Distribution of euclidean distances for points on $\mathcal{P}$ relative to $r^*$ for a given instance of size 600 with EAS-cluster.

(c) Distribution of euclidean distances for points on $\mathcal{P}$ relative to $r^*$ for a given instance of size 600 with EAS-cluster.

Figure 6: Histograms of Euclidean distances of Pareto Front from EAS-cluster compared to NoEAS.

## C    Hyper-volume Additional Statistics

In this section, we report additional statistics on the percentage increase in hyper-volume for our experiments. Our statistics concern the quantity:

$$\frac{[HV(\mathcal{P}^b)_{EAS-cluster} - HV(\mathcal{P}^b)_l] \times 100}{HV(\mathcal{P}^b)_l} \tag{7}$$

for which the mean across all $B$ instances represents the metric $Z$ used in Section 5.2. In addition to the mean, we report the standard deviation, the min and max values in the following Tables. Table 5 concerns results in Table 1, Table 6 concerns results in Table 2 and Table 7 concerns results in Table 3.

| n | Metric | EAS-basic | NoEAS | LGA | LGA+noML | LALNS |
|---|--------|-----------|-------|-----|----------|-------|
| 50 | St Dev | 3.72% | 6.4% | 112.82% | 65.11% | 8.28% |
| | Max | 7.54% | 25.23% | 574.86% | 466.55% | 33.43% |
| | Min | -6.98% | -3.94% | 178.75% | 221.02% | -0.08% |
| 100 | St Dev | 3.97 % | 4.63 % | 206.67% | 147.31% | 6.13% |
| | Max | 10.16 % | 15.8 % | 1251.36 % | 938.33% | 27.53% |
| | Min | -5.63 % | 0.57 % | 348.12 % | 389.11% | 4.10% |
| 200 | St Dev | 2.37 % | 3.34 % | 171.33 % | 241.78% | —— |
| | Max | 5.44 % | 11.15 % | 1372.69 % | 1643.61% | —— |
| | Min | -5.0 % | -0.6 % | 676.73% | 697.78% | —— |

Table 5: Statistics on percentage increase in hyper-volume of EAS-cluster relative to baseline for the experiments reported in Table 1

| n | Metric | EAS-cluster (Greedy) |
|---|--------|----------------------|
| 50 | St Dev | 2.4 % |
| | Max | 6.86 % |
| | Min | -3.87 % |
| 100 | St Dev | 1.95 % |
| | Max | 3.47 % |
| | Min | -5.73 % |

Table 6: Statistics on percentage increase in hyper-volume of EAS-cluster with Monte Carlo Simulation relative to Greedy policy for the experiments reported in Table 2

| n | Metric | No-EAS |
|---|--------|--------|
| 50 | St Dev | 6.57 % |
| | Max | 25.7 % |
| | Min | -2.71 % |
| 100 | St Dev | 4.73 % |
| | Max | 16.66 % |
| | Min | 0.3 % |
| 200 | St Dev | 3.53 % |
| | Max | 10.99 % |
| | Min | -0.87 % |

Table 7: Statistics on percentage increase in hyper-volume of EAS-cluster relative to NoEAS for the experiments reported in Table 3 with more variable travel times.

