# OpenReview forum: "End-to-end Deep Reinforcement Learning for Stochastic Multi-objective Optimization in C-VRPTW"
_TMLR — Accepted by TMLR_

### Review · Reviewer_gJMp · 2025-12-22

**Summary Of Contributions:**

This work introduces a two tiered approach for the stochastic nature of the Vehicle Routing Problem with Time Windows. The first tier is an adaptation of the POMO approach to generate solutions in deterministic settings with preference vectors for the objectives. The second tier applies the Efficient Active Search (EAS) strategy to deal with the stochastic nature of the travel times. To prevent the combinatorial explosion of evaluation across many stochastic scenarios, this work introduces a clustering-based scenario selection approach, by clustering scenarios together if they demonstrate a high similarity in objective values.

Other: The authors claim that this is a 'Regular submission (no more than 12 pages of main content)'. However, this submission has 14 pages of content.

**Audience:**

Yes

**Audience Explanation:**

Yes, I believe this work is relevant to the greater TMLR community as it provides an interesting take on the stochastic nature of the vehicle routing problem. This work sits at the intersection of machine learning, reinforcement learning, optimization and graph theory.

The findings of this work are empirical in nature. Through a series of experiments, the authors demonstrate how EAS clustering improves solution quality under stochastic travel time conditions, which is an interesting take on problems that can easily get intractable.

I believe these findings would be relevant for the interests of the broader TMLR community.

**Broader Impact Concerns:**

A broader impact statement is not present and not required.

**Claims And Evidence:**

Yes

**Claims Explanation:**

The authors describe the two tiered approach to generating solutions to the vehicle routing problem. The clustering technique, the active search and the sampling approach is written in detail along with algorithmic details.
1. The authors conduct numerical experiments on three levels of instances with sizes 50, 100, and 200 with 500 difference stochastic realizations. While this provides a basis for extensive evaluation, the clustering significantly reduces the number of scenarios, and may also harm the quality of the Pareto Front.
2. I found the nature of the baselines weak and lacking. It would be interesting to see how the proposed approach fares against other baselines that are not POMO.
3. Additionally, to convincingly state the strong generalization of this approach to unseen stochastic travel time, more experiments are needed with more larger number of nodes, and a greater degree of standard deviation.

**Requested Changes:**

1. The biggest critique to this work is the lack of baselines. I would be interested in seeing how the proposed approach fares against other baselines that are not POMO.
2. The stochasticity follows a normal distribution with a standard deviation of 20 percent of the expected value. This value is realistic. However, an ablation study along various increasing levels of standard deviation and increasing number of nodes would be incredibly useful in understanding the true benefit of this work.
3. It seems that the true benefit of the EAS-cluster over the non-cluster version seems to be in the time efficiency. The benefit in solution quality does not seem significant. A further explanation along with a statistical significance test would help in understanding the difference between the two approaches.
4. I believe the writing can be improved at several instances. For e.g., 'independently constructs solution without interference of a heuristic' is missing an article. I would recommend taking another pass at the paper to fixing typos and write sentences in a simplified manner.

Other:

1. The authors claim that this is a 'Regular submission (no more than 12 pages of main content)'. However, this submission has 14 pages of content. This discrepancy needs to be fixed.
2. Figures 1 and 2 and subsequently the images in the appendix are bitmap images. They are already pixelated and do not adhere to the standards of TMLR. I recommend using vector graphics for images.
3. Formatting near section 4 seems to have a large empty space.

---

### Review · Reviewer_qeJU · 2025-12-30

**Summary Of Contributions:**

The authors propose slight modifications to existing methodologies (mostly, they start off the one of Lin & al., 2022 and add scenario clustering) that improve their running times.

**Audience:**

Yes

**Audience Explanation:**

The topic of of deep reinforcement learning is of interest to the TMLR community.

**Broader Impact Concerns:**

Not present and not useful for this submission.

**Claims And Evidence:**

No

**Claims Explanation:**

Apart from a minor point (minor comment below), the numerical results do not include comparisons to strong baselines, both in terms of computing times and solution quality. While the authors improve compared to other ML-based techniques, as a potential implementer of the said methodology, I do not know if I should pursue an ML-based implementation or a classical implementation.

**Requested Changes:**

End-to-end Deep Reinforcement Learning for Stochastic Multi-objective Optimization in C-VRPTW

Major comments:
- Section 3, page 3: the authors seem to focus on one method to compute the Pareto front, namely the weighted sum method. This is just one method of producing the Pareto front; in particular, it cannot produce a nonconvex front (as all linearisation methods). As your problem is equivalent to a mixed-integer linear program, the front can be nonconvex. See, for instance, https://pubs.acs.org/doi/10.1021/acs.iecr.1c01175. I think this issue requires some discussion to show the limits of your methodology.
- The considered instance sizes are rather small (up to 200 locations), although the number of scenarios is large.
- Section 5.2: you only compare your methodology to ML-based techniques to solve the problem. I have doubts that you are comparing your technique to a strong baseline. I would prefer to see, among your baselines, a traditional solver for CVRPTW: for instance, LGA is based on eugenic algorithms with some ML, there is no reason why the same eugenic algorithms wouldn't work on the same problem (or other local-search-based or column-generation-based solvers); you could also use the simple weight technique over a stochastic solver to get this baseline.

Minor comment:
- Section 1, page 2: "Although the training resources required are not trivial, ML models can be quickly applied for solving thereafter (Zhang et al., 2021)". The reference only applies to toy problems (TDTSP with up to 100 locations). They get good results, but they are very inconclusive due to the size of the instances. The authors of the submission thus make a very bold claim that is not substantiated by their citation.

Open-ended question (I'd like to see it answered in this paper, but you can delay it to a future paper):
- You consider only two objective functions, which is a very limited case of multiobjective optimisation. It would be interesting to see how your technique generalises to more objectives.

---

### Review · Reviewer_N8t7 · 2026-01-05

**Summary Of Contributions:**

The paper proposes a reinforcement learning-based approach capable of jointly handling stochastic travel times and multiple objectives in the Capacitated Vehicle Routing Problem with Time Windows (C‑VRPTW).
Building on the POMO method, the authors introduce an active search retraining mechanism to efficiently adapt the model’s embedding to uncertainty in travel times. Numerical experiments across multiple instance sizes show that the proposed approach achieves Pareto fronts close to (slightly worse than) alternative active learning protocols, while significantly reducing the computational time. The ablation study confirms the effectiveness of the proposed active search, as the multi-objective POMO alone leads to inferior results.

The model generalizes reasonably well to travel time distributions different from the training ones. The authors tested the case of the variance being doubled with respect to the training distribution.

**Audience:**

Yes

**Audience Explanation:**

The paper focuses on the C-VRPTW. Readers already familiar with this topic would find in the paper a nice technical description of the proposed approach. I think the problem in itself can be of interest for machine learning practitioners (e.g., also related to graph neural networks).

However, as I suggest to the authors, the paper is difficult to read by a broader machine learning audience. If that is not intended, then the paper requires some restructuring and a gentle introduction to the problem at hand, especially when it comes to the evaluation metrics and how to interpret them.

**Broader Impact Concerns:**

No concerns.

**Claims And Evidence:**

Yes

**Claims Explanation:**

The proposed approach effectively employs reinforcement and active learning techniques for the C-VRPTW. The experiments prove the soundness of the design. I am not an expert in the specific problem, therefore I find it difficult to thoroughly evaluate the empirical analysis. However, I did not find any major error on the reinforcement or active learning side of the approach.

**Requested Changes:**

- The current version of the paper is very technical and interesting for a reader already very familiar with the C-VRPTW. To make the paper more accessible to a wider audience, I would suggest providing a gentle introduction to the problem. In particular, the empirical evaluation is quite difficult to understand. To support the formal description of the evaluation metrics, I suggest discussing how to interpret them, for example, by providing an intuitive understanding of the upper/lower bounds for each metric. Overall, a more detailed description of the experimental protocol and results would really benefit the reader interested in the problem but not well-versed in its technical details. I expect most of TMLR readers to fall in this latter category.

- The paper could have provided a robustness analysis on some of the design choices that are not explicitly addressed in the current version. For example, the choice of $\epsilon=1\%$ is provided with no explanation. Did some tuning of $\epsilon$ occur? Or is the method robust to variations of its value?

---

### Comment · Action_Editor_7Xgw · 2026-03-05
**Camera (not) ready**

Dear authors,

the camera ready revision is still anonymized.

You should replace \usepackage{tmlr} with \usepackage[accepted]{tmlr}, insert all author information, and the link to the OpenReview page for the submission.

Best,

AE

---

> ### Author Response · Authors · 2026-03-05
> **Camera (hopefully) ready**
>
> Dear AE,
>
> The changes were accordingly added.
>
> Regards,
> Authors

---

> > ### Author Response · Authors · 2026-03-11
> > **Publication Date**
> >
> > Dear AE,
> >
> > Is there a predicted timeline for the publication of our paper? It would be helpful if its not delayed too much.
> >
> > Regards,
> > Authors

---

### Decision · Action_Editor_7Xgw · 2026-02-24

**Recommendation:** Accept as is

**Audience:**

Yes

**Audience Explanation:**

Initially, there were some concerns on the readability of the manuscript, especially for a general machine learning audience less familiar with the specifics of vehicle routing.

Some parts of the paper were rewritten and the current revision is accessible to a broader audience.

**Claims And Evidence:**

Yes

**Claims Explanation:**

Initially, there were some concerns regarding the size of the problems considered in the experiments, the lack of traditional baselines, and the strength of claims regarding ML-based techniques.

The authors addressed these issues either by adding more experiments or by editing the claims.

The current revision convincingly supports the claims.